# Ultrasound as a reliable guide for lumbar intrathecal injection in rats: A pilot study

Pryambodho[1,2¤]*, Ismail Hadisoebroto Dilogo[3☯¤], Aida Rosita Tantri[2☯¤], Renindra Ananda Aman[4☯¤], Tjokorda Gde Agung Senapathi[5☯], Jan Sudir Purba[6☯¤], Nuryati Chairani Siregar[7☯¤]

1 Doctoral Program in Medical Sciences, Faculty of Medicine, Universitas Indonesia, Jakarta, Indonesia, 2 Department of Anesthesiology and Intensive Care, Faculty of Medicine Universitas Indonesia, Cipto Mangunkusumo National Central Hospital,ee Jakarta, Indonesia, 3 Department of Orthopedic and Traumatology, Faculty of Medicine Universitas Indonesia, Cipto Mangunkusumo National Central Hospital, Jakarta, Indonesia, 4 Department of Neurosurgery, Faculty of Medicine Universitas Indonesia, Cipto Mangunkusumo National Central Hospital, Jakarta, Indonesia, 5 Department of Anesthesiology and Intensive Care, Faculty of Medicine Udayana University, Bali, Indonesia, 6 Department of Neurology, Faculty of Medicine Universitas Indonesia, Cipto Mangunkusumo National Central Hospital, Jakarta, Indonesia, 7 Department of Pathological Anatomy, Faculty of Medicine Universitas Indonesia, Cipto Mangunkusumo National Central Hospital, Jakarta, Indonesia

☯ These authors contributed equally to this work.
¤ Current address: Department of Anesthesiology and Intensive Care, Universitas Indonesia, Cipto Mangunkusumo National Central Hospital, Jakarta, Indonesia
* pry@cbn.net.id

**Data Availability Statement:** All relevant data are within the manuscript and its Supporting Information files We confirm that all raw data required to replicate the study results are available

## Abstract

Lumbar intrathecal administration provides an ideal route for drug delivery into the central nervous system, especially when dorsal root ganglions are the main target for the therapy in rat model of chronic pain. Two main methods of lumbar intrathecal administrations are chronic catheter implantation and the acute needle puncture. Chronic catheter implantation involves surgical manipulation to insert micro indwelling catheter into the intrathecal space. However, this method is invasive, produces inflammatory reactions, and generates more surgical stress. Acute needle puncture is less invasive and cheaper however is technically challenging to perform. We performed an ultrasound-guided lumbar intrathecal injection in six male Sprague Dawley rat cadavers, on average weighing 250–300 grams. Fresh rat cadavers were positioned in a sternal recumbent position, vertebrae were palpated and scanned using a linear probe ultrasound. A 25G needle insertion was advanced with real-time ultrasound guidance, and placement was confirmed prior to dye injection (Methylene blue, Sigma Aldrich). Cadavers were then dissected, and the vertebrae were visually inspected for dye staining. All three cadavers that underwent intrathecal injection with sagittal and axial plane ultrasound guidance showed positive dye staining within the intrathecal space, confirming successful acute intrathecal administration. There was one successful intrathecal injection under sagittal plane-only ultrasound guidance. Ultrasound is a useful, operator-dependent tool to guide acute needle puncture intrathecal administration.

within the manuscript, attached figures and
supplementary video file.

**Funding:** The author(s) received no specific
funding for this work.

**Competing interests:** The authors have declared
that no competing interests exist

## Introduction

Intrathecal drug administration is the most common route to deliver drugs into the central
nervous system, overcoming the blood-brain and blood-spinal cord barriers. Two main intra-
thecal administration techniques in animal study are acute needle puncture and chronic cathe-
ter implantation [1]. Acute needle puncture is a simpler and less invasive procedure compared
to chronic catheter implantation procedures, which need surgery and are also costly.

Lumbar intrathecal route is an ideal method when dorsal root ganglions and spinal cord are
the main target for therapy. Lumbar intrathecal acute needle puncture involves using a trans-
cutaneous needle puncture to reach the intrathecal space. Then, following a placement confir-
mation, the needle is connected to a syringe, and injection is performed. This technique is
challenging to perform whilst mostly done only with palpation to identify the targeted interspi-
nous space and presents a particular difficulty in verifying needle position whether correctly in
intrathecal space or not, especially in small experimental animals such as rats and mice. The
most common signs to confirm intrathecal needle position are the tail flick and cerebrospinal
fluid (CSF) flow through the needle [2, 3]. Both signs have their disadvantages. Sudden lateral
tail flick may be diminished in anesthetized rats, which makes this sign difficult to observe and
may not be found frequently. Meanwhile, CSF flow needs a very fine needle to apply and some
mechanical triggers to increase the intrathecal pressure of the rats [3].

Ultrasound (US) has been acknowledged as an indispensable tool for guidance and needle
placement verification in both human and animal subjects. Ultrasound-guided (USG) needle
puncture in humans is now the most reliable technique performed in regional anesthesia for
peripheral nerve block and neuraxial blocks (including intrathecal block). It is a low-cost, mul-
tipurpose tool that allows real-time identification of needle position and anatomical structures.
We theorized that ultrasound might complement the shortcomings of the acute needle punc-
ture technique by reducing the technical difficulty by providing guidance using real-time nee-
dle position, confirmation of the insertion of the needle into the intrathecal space, and image
documentation. Furthermore, this method will be useful in all animal models. Thus, we seek
to establish a novel method of acute intrathecal needle puncture, using ultrasound as a method
for needle guidance and injection placement confirmation.

## Materials and methods

This study did not obtain any ethical clearance specifically for this procedure, since this study
involved rat cadavers which were previously unalived according to ethical standards from
other study with protocol number R.08-22-IR from Animal Care and Use Committee of PT.
Bimana Indomedical.

Total of six male Sprague-Dawley rat cadavers weighing 250–300 grams were included in
this study. The rat cadavers were positioned in the sternal recumbent position, interspinous
space of L6–S1 was identified using Lumify L12-4 linear array transducer ultrasound device
(Lumify, Philips). Intrathecal injections were performed in two different view and approaches
using needle size 25G 0.5 mm x 25 mm (BD PrecisionGlide™ Needle). In the first three cadav-
ers, intrathecal injections were performed under USG in the sagittal view, by firstly place the
probe at the midline and identify the lamina and interlaminar space, then the needle was
advanced through the interspinous space to the targeted intrathecal space using in-plane
approach (Technique 1) (Fig 1), then the cadaver's spine was dissected to document findings.

Based on the first three cadaver findings, three other cadavers' intrathecal injections were per-
formed using different technique approach (Technique 2). Technique 2 started with identifying
the L6–S1 interspinous space using sagittal view, then at the level of the target interspinous space,
USG probe was rotated 90˚ showing transverse view of the interspinous space, then the needle

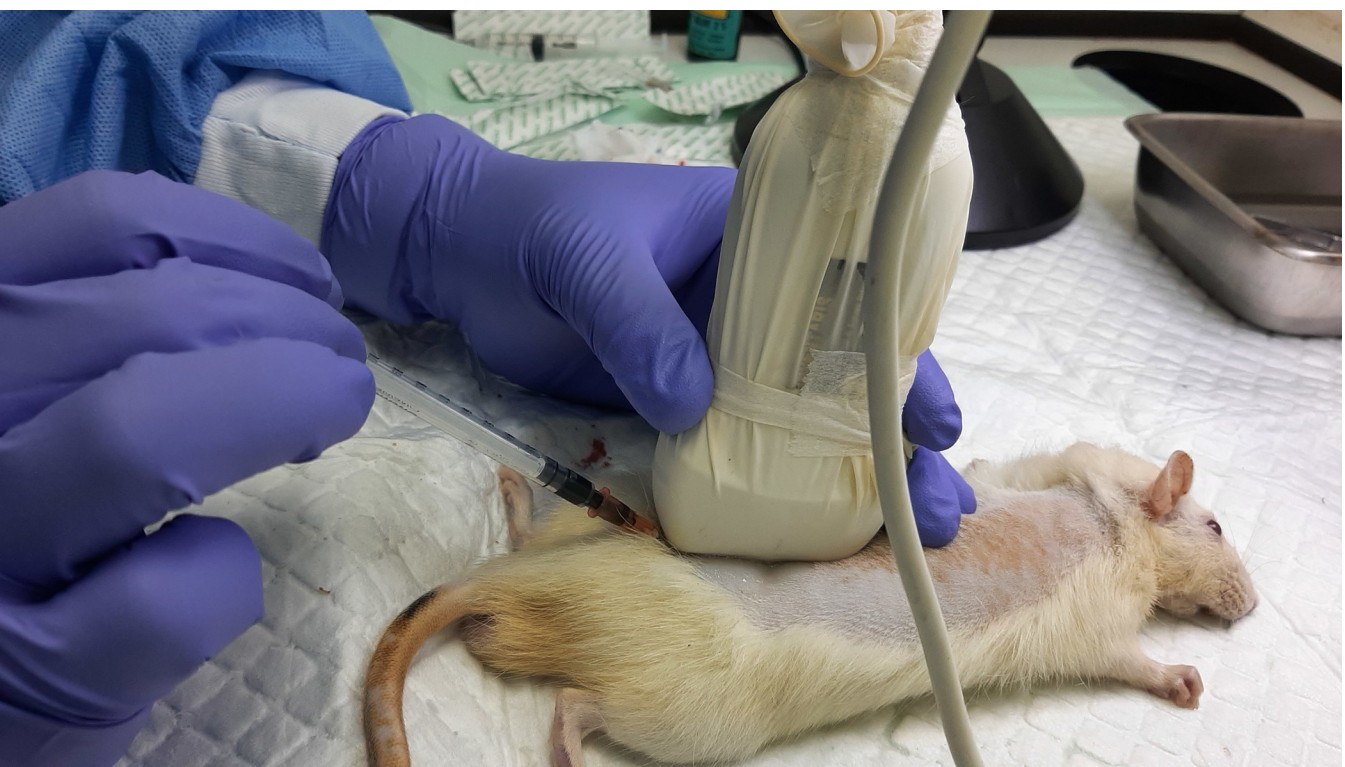

**Fig 1. Technique 1 approach.** Sagittal view using the in-plane approach. Image courtesy from author's database.

was advanced with out-of-plane approach (Fig 2). The second three cadaver's spine injected using Technique 2 was then dissected to document intrathecal injection and dye spreading. All needle insertions were performed under real-time USG, after the needle tip was estimated to be in subarachnoid space, 0.2 ml of dye (Methylene blue, Sigma Aldrich) was injected. The initial USG image obtained before procedure were shown in Fig 3 for Technique 1 and Fig 4 for Technique 2.

## Results

One of three cadaver had successful intrathecal injection using Technique 1 compared to all three cadavers using Technique 2, confirmed by positive dye spreading in the subarachnoid space. Furthermore, there were no false route documented in Technique 2 (Fig 5), compared to the Technique 1 with two out of three cadavers had false dye spreading into paravertebral intramuscular space (Fig 6). Nevertheless, both ultrasound views were able to correctly identify and confirm the interspinous space targeted, also guiding the needle tip into the subarachnoid space and confirmed the placement of the tip within the subarachnoid space (Figs 7 and 8). Ultrasound-guided placement was confirmed by the presence of methylene blue dye within the subarachnoid space. The real-time USG image while performing needle advancement can be viewed in supporting video file (S1 and S2 Files).

## Discussion

### Anatomy

Similar to humans, rats have cervical, thoracic, and lumbar spinal vertebrae. Rat spine consists of 57 vertebrae, distributed as 7 cervical vertebrae, 13 thoracic, 6 lumbar, 4 sacral, and 27

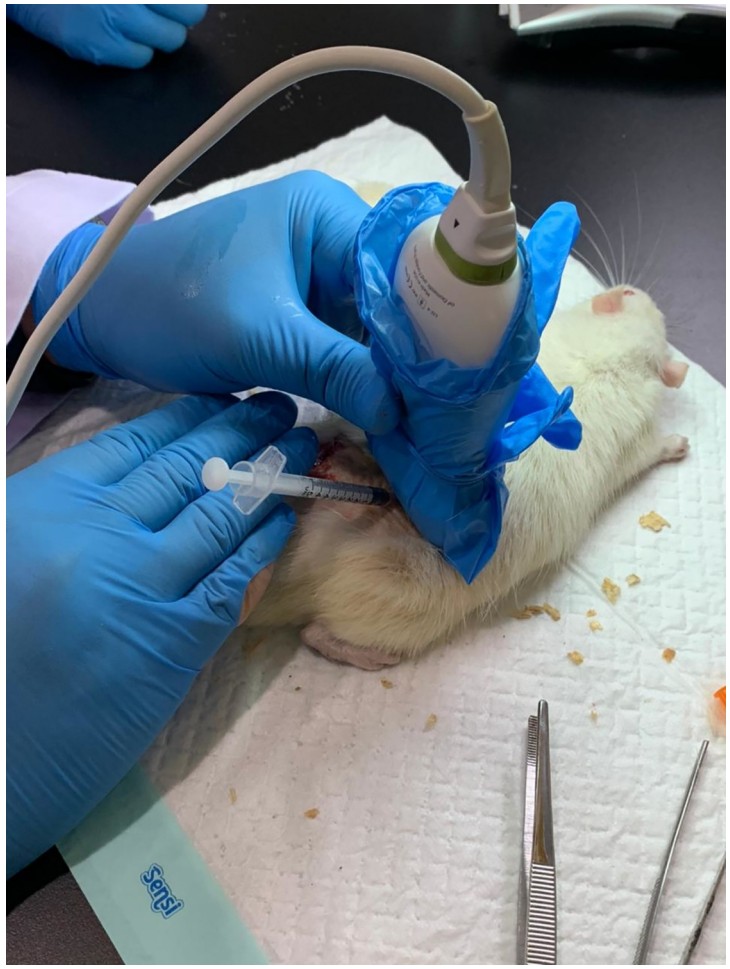

**Fig 2. Technique 2.** Firstly, the target interspinous space was identified using sagittal view, then the probe was rotated 90˚ to transverse view and injection was performed with the out-of-plane approach. Image courtesy from author's database.

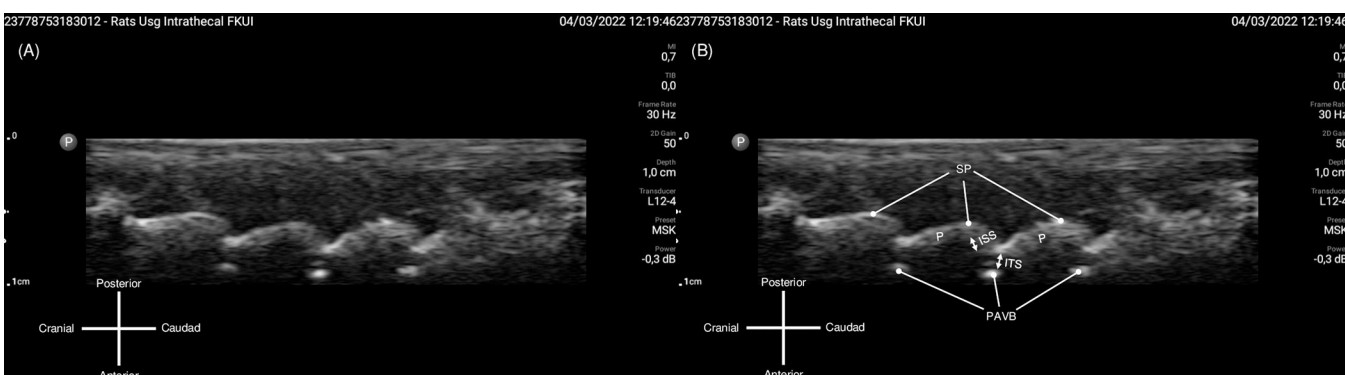

**Fig 3. Sagittal ultrasound view obtained for Technique 1 approach.** (A) Original image obtained from sagittal probe placement. (B) The bony parts of the spine were identified from the hyperechoic rim reflected. ISS: interspinous space, ITS: intrathecal space, PAVB: posterior aspect of vertebral body, P: Pedicle, SP: spinous process. Image courtesy from author's database, captured from USG screen.

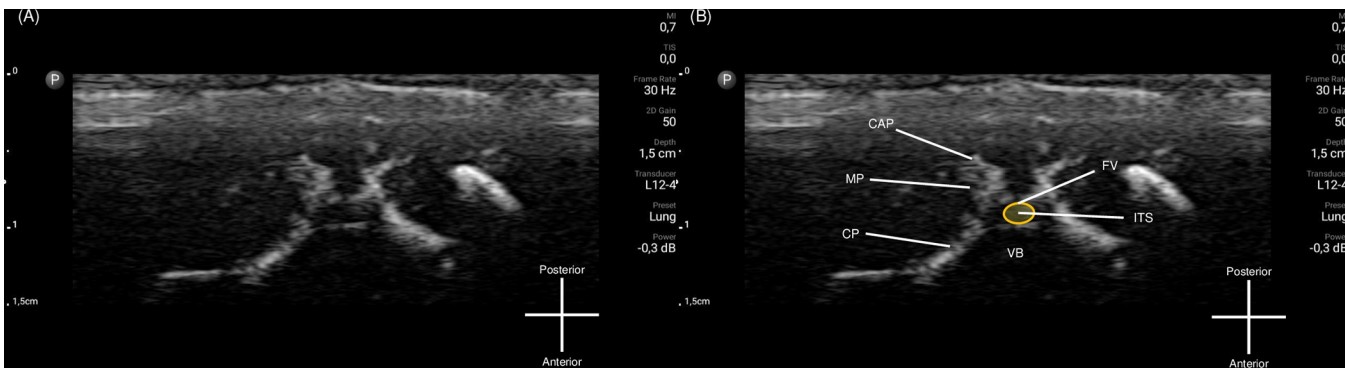

**Fig 4. Transverse ultrasound view for Technique 2 approach.** (A) Original image obtained from transverse probe placement. (B) The bony parts of the spine were identified from the hyperechoic rim reflected. CAP: Caudal articular process, CP: Costal process, FV: Foramen of vertebra (bordered by orange line), ITS: Intrathecal space (area highlighted in yellow), MP: Mamilloarticular process, VB: vertebral body. Image courtesy from author's database, captured from USG screen.

caudal vertebrae. The 13 thoracic vertebrae consist of the body, processes, and arches. It has facets and pedicles relevant to connecting the vertebrae body. Spinal nerves exit from the intervertebral foramen via the intervertebral spaces. The lumbar vertebrae of the rat consist of notches, vertebral bodies, and processes. The lumbar vertebrae are thick and wide and progressively lengthen from cranial to caudal before they shorten again. Spinal cord ends in a conus medullaris (CM), situated at the L3–4 in adult rats then extends to the sacral region as filum

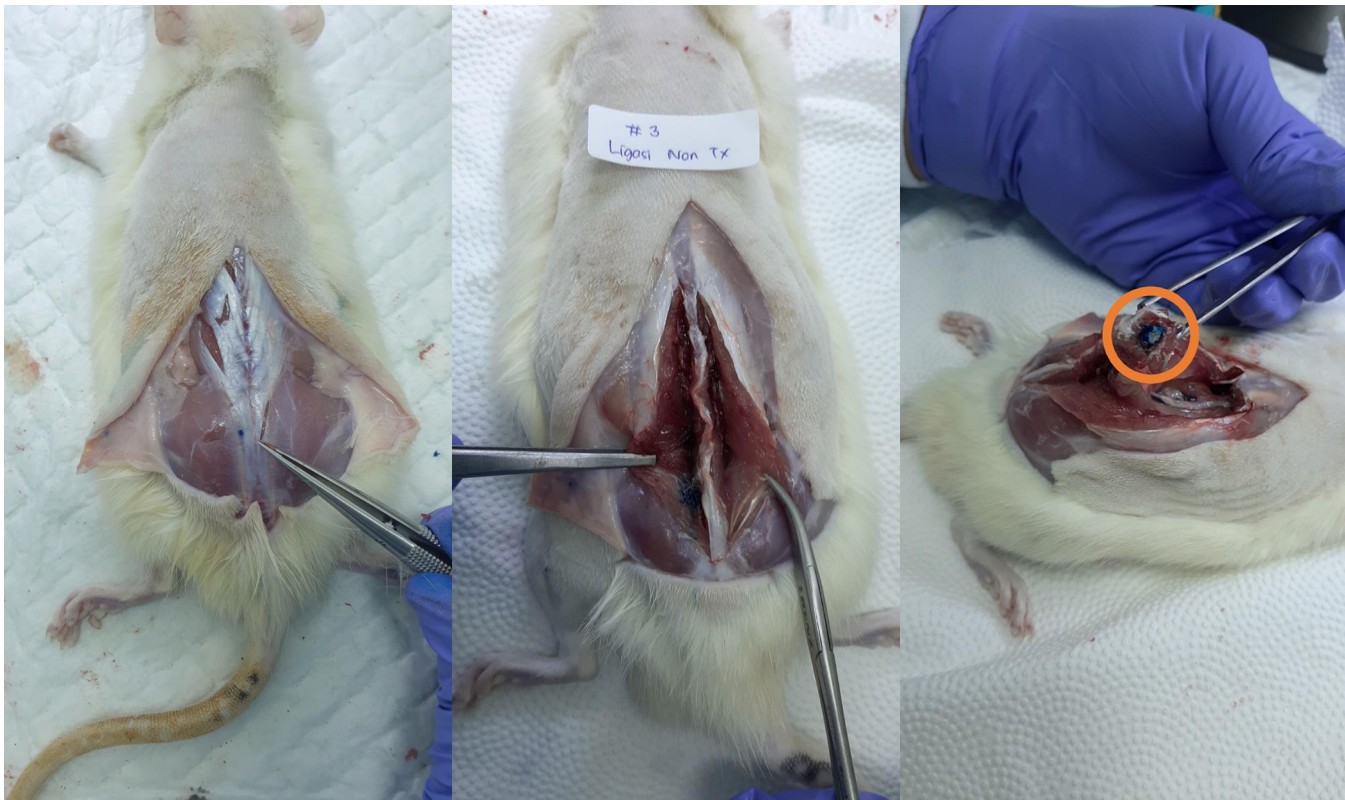

**Fig 5. Dye spreading in all three of cadavers after injection using Technique 2.** Intrathecal space dye spread was shown in orange circle. No dye spreading in erector muscles. Image courtesy from author's database.

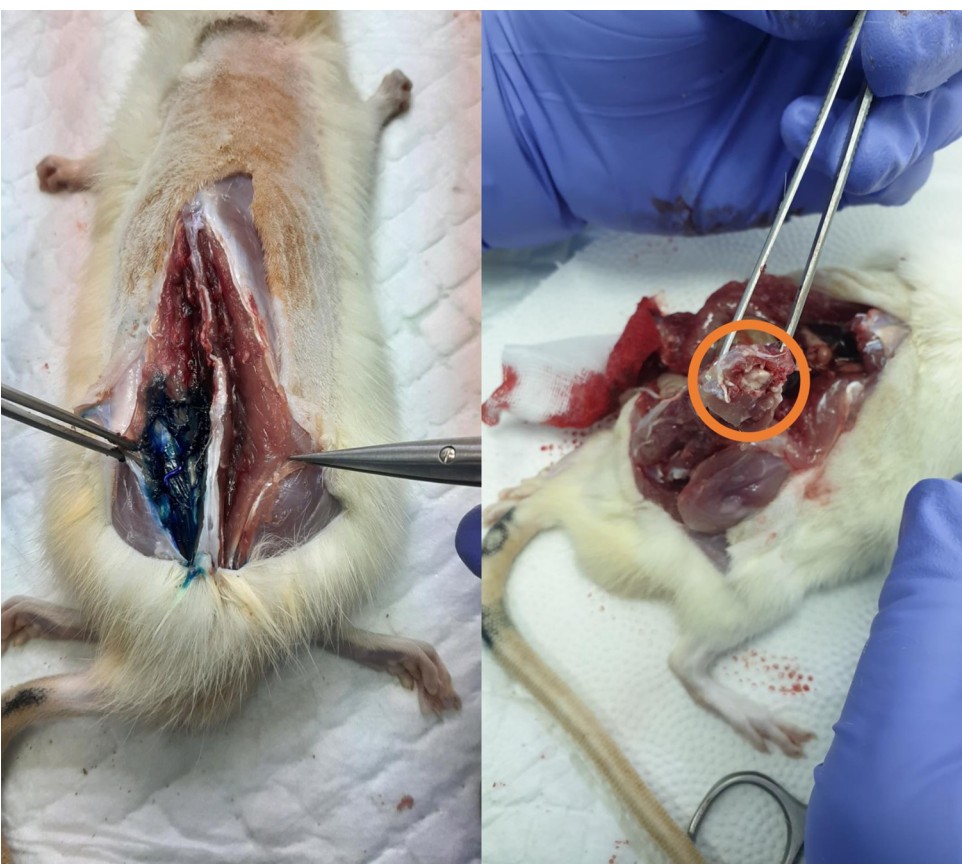

**Fig 6. Dye spreading in erector spine muscles in two of three cadavers with Technique 1.** Dye spread suggesting the unsuccessful intrathecal injection, no dye found intrathecal space shown in orange circle. Image courtesy from author's database.

terminale. Filum terminale (FT) is a thin strand continuous with CM, and tracked caudally 4–5 cm down to the level of No. 5 sacral vertebrae, in contrast the CM is in the form of an inverted cone, becoming thinner caudally owing to gradual decrease of spinal cord

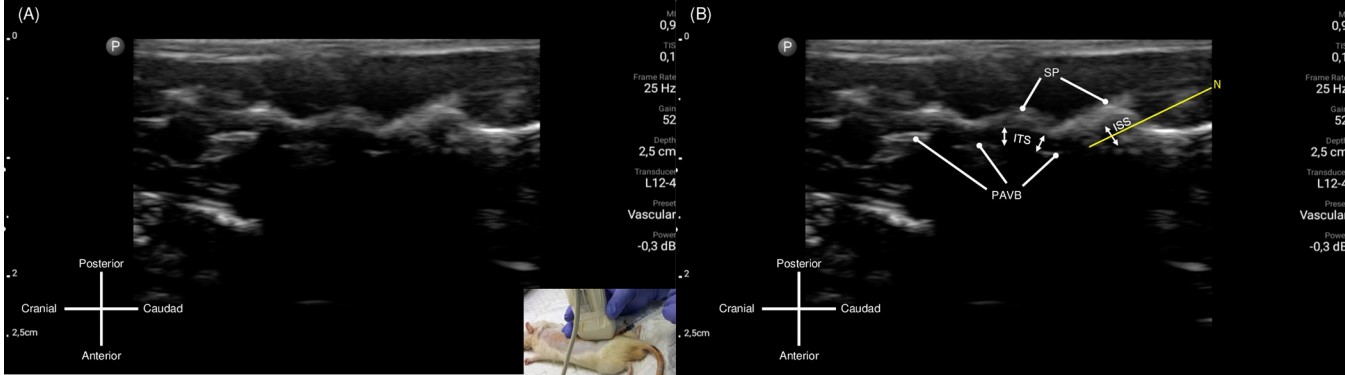

**Fig 7. Needle advancement and placement using Technique 1.** (A) Still image captured while performing needle advancement. Probe placement and needle insertion demonstrated in bottom-right corner of the picture. (B) Needle tip was visualized and advanced with real-time ultrasonography visualization with target of reaching the intrathecal space via interspinous space. ISS: Interspinous space; ITS: Intrathecal space; N: Needle path (augmented by yellow line); PAVB: Posterior aspect of vertebral body; P: Pedicle; SP: Spinous process. Image courtesy from author's database, captured from USG screen.

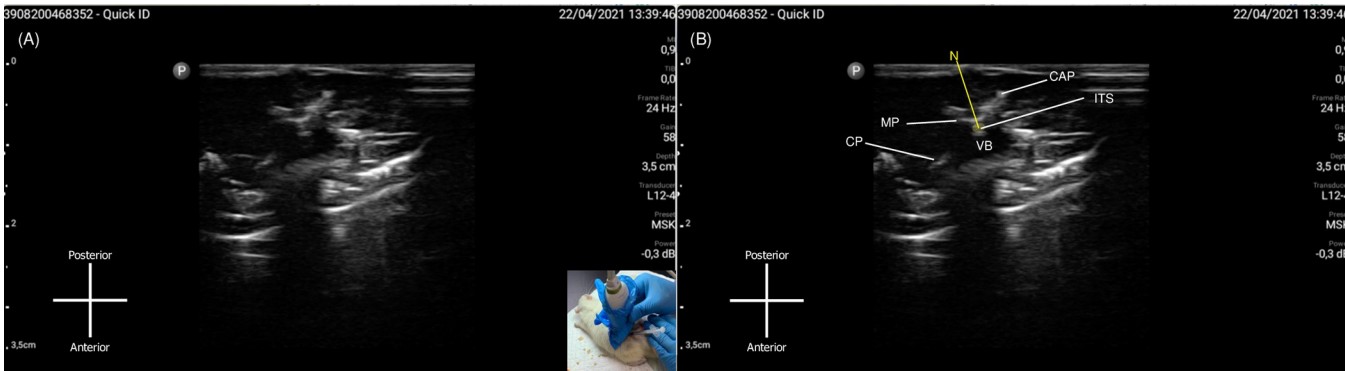

**Fig 8. Needle advancement and placement using Technique 2.** (A) Still image captured while performing needle advancement. Probe placement and needle insertion demonstrated in bottom-right corner of the picture. (B) Needle tip was visualized and advanced with real-time ultrasonography visualization with target of reaching the intrathecal space after identifying interspinous space from the previous sagittal view. CAP: Caudal articular process, CP: Costal process, ITS: Intrathecal space (area highlighted in yellow), MP: Mamilloarticular process, N: Needle path (augmented by yellow line), VB: Vertebral body. Image courtesy from author's database, captured from USG screen.

parenchymal tissue. The CM contains central canal (CC) and an appreciable amount of spinal cord parenchymal tissue, while the FT consists of the CC associated with no, or only a negligible amount of spinal cord parenchymal tissue, making it more safe for spinal puncture [4, 5].

Landmark in intrathecal injection is crucial to identify the correct level of injection. Previously described by Thomas AA, et al. [6] identifications of the lumbosacral region can be achieved by positioning the rats in a sternal recumbent position, maximally bending the vertebrae. In our case, we identified the proposed level of L1–L2 by palpating the back region of the rat, corresponding to the linear region of L4-L5.

## Ultrasound imaging

Ultrasonography involves the emission of sound waves at a certain frequency to identify the relevant structures, with the waves reflected or transmitted to certain surrounding tissues will produce certain level of echogenicity. Bone structures will appear to be black or anechoic on US, with a bright hyperechoic rim, and due to the inability of the beam to penetrate the bone, it casts an acoustic shadow beyond it. By using the different densities of anatomical structures, one can identify the internal anatomy without performing any dissection. Generally, the bony vertebrae and the vertebral foramen can be easily identified as anechoic structures with hyperechoic rim and acoustic shadow underneath. As the needle passes through the structure and reaches the intrathecal space, the tip of the needle, visible on the ultrasound as hyperechoic, should be apparent, penetrating through the tissue towards the acoustic shadow. With knowledge of the anatomical shape, we can identify the type of vertebrae of the structure. Proper identification with the relevant clinical anatomical landmark will allow users to infer the level of the spinal column [7].

Despite the clinical usefulness, a certain level of skill set is required. Proper training and adequate experience will yield better results as ultrasound is known to be operator-dependent. Nonetheless, with appropriate handling and needling, ultrasound is a valuable asset for needle injection and confirmation for needle placement. In this study, following the identification of the proper level of the spinal column, the rat was scanned using a linear probe to identify the shape of the vertebrae and the foramen, corresponding to the intrathecal space. The needle was then advanced into the intrathecal space, using ultrasound to identify the position of the tip of the needle. Once the intrathecal placement was confirmed, the dye was then injected.

Real-time ultrasonography guide in any procedure was highly dependent on the operator experiences and dexterity. Technique 1 provides overview of the spine and other interspinous spaces. This technique also could provide the whole needle view and its progression during insertion and advancement. However, given the acute angle and relatively small and short (25G) needle, the needle appearance was rather vague thus making it more challenging to determine the tip of the needle. Technique 2 used in this study provided a short axis view of the targeted interspinous space, operator would need to estimate the tip of the needle, however it could be more advantageous in an acute-angle approach.

## Other mechanisms of administration

The intrathecal injection in mice was first performed by Yaksh and Rudy in 1976 via a catheter, but their method is complicated and unsuitable for a single injection. In 1980, Hylden and Wilcox described a method to reliably perform intrathecal injection in mice [2, 8]. Akerman adapted the Hylden and Wilcox technique and inserted a catheter into rats' subarachnoid space in 1985, yielding a success rate of 80% (8 of 10 rats) [8]. In 1994, Mestre and colleagues adapted this method to rats; they performed intrathecal injection to dorsal aspects of L5 and L6 blindly, and sudden lateral tail movement was observed when the needle was used to indicate needle entry to subarachnoid space. There were instances of false injection to paravertebral muscle; the number of failed injections was not mentioned, however [2].

The other method is performing intrathecal injection on anesthetized mice mounted on a specially designed setup under ultrasound guidance and confirmed by gadolinium-based magnetic resonance imaging (MRI). Choi and colleagues performed this injection on L5–L6, identified using ultrasound, and yielded a success rate of 80% (16 out of 20) [9].

In comparison to indwelling catheters, acute needle puncture has its benefits. Firstly, it is cheap and fast, without needing to perform the time-consuming laminectomy. It does not impose surgical stress on the study models and does not need the expensive indwelling catheter [1]. Placement of a catheter also induces immune reactions and has risk for spinal cord trauma, developing idiopathic hemorrhage or abscesses [10]. For single-shot injection, acute needle puncture is ideal, provided that tip placement can be confirmed.

For confirmation, tail flick and CSF tap are usually reported in experiments. However, any displacement during injection might not be detected in some subjects. Due to technical challenges requiring restraint methods like anesthesia, multiple puncture attempts might be required before successful insertion without any imaging aid, other drawback include a significant chance of damaging spinal or nerve roots. To address these challenges, imaging modality had been demonstrated in rodent models by Pasarikovski et al. using real-time intraoperative imaging guidance like c-arm fluoroscopy [10, 11]. The use of ultrasound demonstrated in this study provides visual guidance for needle puncture, reducing the number of attempts and providing evidences of successful intrathecal needle placement by visually ensuring that the needle tip remains in place during intrathecal administration.

## Study limitations

The limitation present in this study is that we used cadaveric rat models, which might ease the level of skillset for USG scanning. However, in anesthetized rat models, we assumed that the mobility of the rat would be largely impaired, similar to cadaver models. We were also unable to assume the extent of the spread of the dye using a cadaveric model. We also had a small number of subjects, as this is a feasibility study and a larger study with a living rat model is planned to take place in the near future. Larger study with larger sample size would give a more appropriate validity and better comparison between techniques.

## Conclusions

Ultrasound proved to be a sound and accurate tool for tip placement confirmation for acute needle injection in rats for trained operators. Both single sagittal view approach and transverse view after sagittal view could be used to perform intrathecal injections in rats. Transverse and sagittal view demonstrated a higher probability in intrathecal injection success. However, this finding is operator-dependent and needed further study to compare success rate. Further study involving living rat model and larger sample group is required to confirm the usefulness and effectiveness of these techniques. We recommend this technique to be performed with an operator which already familiar in using ultrasonography in various procedure involving needle puncture. Also, consider in using smaller USG probe than standard linear probe, such as small footprint, for better image identification and more precise estimation of the needle relative to the anatomical bony landmarks.

## Supporting information

**S1 File. Video documentation of USG view while performing Technique 1.** Anatomy identification refer to Fig 7.
(MP4)

**S2 File. Video documentation of USG view while performing Technique 2.** Anatomy identification refer to Fig 8.
(MP4)

## Acknowledgments

We thank Fitriya N. A. Dewi, DVM, Ph.D, Cert., L.A.M. and Permanawati, DVM, Cert. DHPHL for their help in handling the rat cadavers during the process of this study. We also thank Erika Sasha Adiwongso, MD for her help during the publication process.

## Author Contributions

**Conceptualization:** Pryambodho, Ismail Hadisoebroto Dilogo, Renindra Ananda Aman, Tjokorda Gde Agung Senapathi, Jan Sudir Purba.

**Methodology:** Pryambodho, Aida Rosita Tantri.

**Resources:** Pryambodho.

**Supervision:** Ismail Hadisoebroto Dilogo, Aida Rosita Tantri, Renindra Ananda Aman, Tjokorda Gde Agung Senapathi, Jan Sudir Purba, Nuryati Chairani Siregar.

**Validation:** Pryambodho.

**Writing – original draft:** Pryambodho.

**Writing – review & editing:** Pryambodho, Aida Rosita Tantri.

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
