## [Decision Letter · Decision Letter 0]

18 Apr 2024

PONE-D-24-10769Ultrasound as a reliable guide for lumbar intrathecal injection in rats: A pilot study

PLOS ONE

Dear Dr. Pryambodho Pryambodho, 

Thank you for submitting your manuscript to PLOS ONE. After careful consideration, we feel that it has merit but does not fully meet PLOS ONE’s publication criteria as it currently stands. Therefore, we invite you to submit a revised version of the manuscript that addresses the points raised during the review process.

*Good idea, but we suggest some corrections,*

Your manuscript sample size of the study is very small,

*The use of paramedian oblique sagittal (PMOS) plane can increase visibility of the posterior complex, including the ligamentum flavum and posterior dura,The needle orgin, length and ultrasound image can be added.*

We look forward to receiving your revised manuscript.

Kind regards,

Hilmi Demirkiran

Academic Editor

PLOS ONE

Journal Requirements:

2. We note that your Data Availability Statement is currently as follows: [All relevant data are within the manuscript and its Supporting Information files]

Additional Editor Comments (if provided):

Good ideabut some corrections necessary

Reviewers' comments:

Reviewer's Responses to Questions

**Comments to the Author**

1. Is the manuscript technically sound, and do the data support the conclusions?

Reviewer #1: Yes

Reviewer #2: Yes

Reviewer #3: Yes

2. Has the statistical analysis been performed appropriately and rigorously? 

Reviewer #1: N/A

Reviewer #2: N/A

Reviewer #3: N/A

3. Have the authors made all data underlying the findings in their manuscript fully available?

Reviewer #1: Yes

Reviewer #2: No

Reviewer #3: Yes

4. Is the manuscript presented in an intelligible fashion and written in standard English?

Reviewer #1: Yes

Reviewer #2: Yes

Reviewer #3: Yes

5. Review Comments to the Author

Reviewer #1: Dear author,

I reviewed your study titled "Ultrasound as a reliable guide for lumbar intrathecal injection in rats: A pilot study" with interest. I've added some suggestions and comments below.

Recommendation 1: The origin of the needle, its length, and its visibility on ultrasound should be stated.

Recommendation 2: Specifying the level where the spinal cord ends in the Discussion-Anatomy section provides more descriptive information for a safe intrathecal injection site.

Recommendation 3: Technique 1: Wouldn't it be more appropriate to express it as "paramedian sagittal" instead of "longitudinal-sagittal"?

Comment 1: While 3 out of 3 successes were reported in technique 2, 1 out of 3 success was reported in technique 1. The use of the paramedian oblique sagittal (PMOS) plane could increase the chance of success by providing visibility of the posterior complex including the ligamentum flavum and posterior dura.

Comment 2: The importance of such studies will increase as preliminary examination and/or real time ultrasound will be used to reduce technical difficulties in spinal anesthesia applications with anatomical landmarks in humans, and ultrasound image recording will be evidence in possible legal/judicial procedures.

Reviewer #2: Suggested minor corrections have been indicated on the text, these corrections should be taken into consideration.

Conclusion section of the mansucript may be extended. Some suggestion for future may be added to the conclusion section of the manuscript

Reviewer #3: As you mentioned in your article sample size of the study is very small. But if your study made larger, its validity may become more appropriate. But you also mentioned the small sample size in your study. I recommend planning a study with a sufficiently large sample size.

6. PLOS authors have the option to publish the peer review history of their article (what does this mean?). If published, this will include your full peer review and any attached files.

Reviewer #1: **Yes: **Mehmet Selim ÇÖMEZ

Reviewer #2: No

Reviewer #3: No

---

## [Author Response · Author response to Decision Letter 0]

30 May 2024

Thank you for your response and input for our manuscript entitled “Ultrasound as a reliable guide for lumbar intrathecal injection in rats: A pilot study” and considered for publishing by PLOS One.

We would like to address some of the queries from reviewers

Reviewer #1

Recommendation 1: The origin of the needle, its length, and its visibility on ultrasound should be stated.

We have added additional information regarding the origin and length of the needle in the manuscript, specifically line 65–66, needle visibility on ultrasound was described at line 175–8. Thank you for the recommendation

Recommendation 2: Specifying the level where the spinal cord ends in the Discussion-Anatomy section provides more descriptive information for a safe intrathecal injection site.

We have added the information regarding where the spinal cord ends in the anatomy section, specifically line 150–8, thank you for the input.

Recommendation 3: 

Technique 1: Wouldn't it be more appropriate to express it as "paramedian sagittal" instead of "longitudinal-sagittal"?

We agree with the term technically, however given the large probe compared to the tiny vertebra of the rat, we weren’t sure of the exact plane shown was true perpendicular median-sagittal or slightly paramedian-sagittal oblique. We agree to change the terminology to a more general term in describing the probe position, sagittal view. 

Comment 1: While 3 out of 3 successes were reported in technique 2, 1 out of 3 success was reported in technique 1. The use of the paramedian oblique sagittal (PMOS) plane could increase the chance of success by providing visibility of the posterior complex including the ligamentum flavum and posterior dura.

We agree that PMOS plane was shown to augment posterior complex in human, however it is difficult to determine in rats. Slight adjustment with a large probe compared to the tiny vertebra, it is mathematically difficult to determine whether the plane shown was true median/paramedian. For note, the distance between midline to the edge of lamina was approximately less than 0.5 mm in rats. It required dynamic scanning to obtained the documented view, with recommendation of firstly place the probe at the midline, then identify the lamina and interlaminar space. (Line 66¬–70). The probe would be in sagittal plane between the median to paramedian line, facing the interspinous space. Therefore we also recommend this to be performed by an operator which already familiar in using USG in various procedure involving needle puncture which we added in line 246–8.

Comment 2: The importance of such studies will increase as preliminary examination and/or real time ultrasound will be used to reduce technical difficulties in spinal anesthesia applications with anatomical landmarks in humans, and ultrasound image recording will be evidence in possible legal/judicial procedures.

The direction of this study were not as a support for preliminary examination and/or real time ultrasound for reducing spinal anesthesia anatomical landmarks and technical difficulties in humans, since the ultrasound uses and advantages in human, especially during neuraxial procedure, were quite established. We rather had the inspiration from that experience for similar application in rats since ultrasound aid had not been demonstrated before and was more challenging to perform in rats. We hoped that this method could give a more feasible, reliable, and objective guide compared to the previous traditional tail-flick method. 

Reviewer #2: 

Suggested minor corrections have been indicated on the text, these corrections should be taken into consideration.

We have reviewed the corrections indicated in the text, thank you for your suggestions

Conclusion section of the manuscript may be extended. Some suggestion for future may be added to the conclusion section of the manuscript

Thank you for the input, we have added suggestion regarding future studies in line 251–7.

We would also like to address the concern of reviewer 2’s answer regarding the editorial question of ‘Have the authors made all data underlying the findings in their manuscript fully available?’ Which data underlying the findings weren’t available according to reviewer 2 so that we could improve?

Reviewer #3

As you mentioned in your article sample size of the study is very small. But if your study made larger, its validity may become more appropriate. But you also mentioned the small sample size in your study. I recommend planning a study with a sufficiently large sample size.

We agree and aware of the issue regarding the small sample size in this current small study, but for a pilot study and technique demonstration purpose, we hope that this study could show the feasibility of the proposed technique (Line 242–3). In the near future, we planned to publish our study which utilize this technique, with larger sample size and thus better validity, with a main purpose of therapeutic intrathecal injection in neuropathic rats model. 

We thank the reviewers for their time and input regarding this manuscript. Please address further queries concerning this manuscript to Pryambodho at pry@cbn.net.id. We look forward to hear from you at your earliest convenience. Thank you for your consideration and attention on this manuscript.

---

## [Decision Letter · Decision Letter 1]

22 Aug 2024

Ultrasound as a reliable guide for lumbar intrathecal injection in rats: A pilot study

PONE-D-24-10769R1

Dear Dr. Pryambodho,

We’re pleased to inform you that your manuscript has been judged scientifically suitable for publication and will be formally accepted for publication once it meets all outstanding technical requirements.

Kind regards,

Ramada Rateb Khasawneh

Academic Editor

PLOS ONE

Additional Editor Comments (optional):

thank you for submitting your manuscript to Plos one

It looks good

Good Luck

Reviewers' comments:

Reviewer's Responses to Questions

**Comments to the Author**

1. If the authors have adequately addressed your comments raised in a previous round of review and you feel that this manuscript is now acceptable for publication, you may indicate that here to bypass the “Comments to the Author” section, enter your conflict of interest statement in the “Confidential to Editor” section, and submit your "Accept" recommendation.

Reviewer #1: All comments have been addressed

Reviewer #4: All comments have been addressed

2. Is the manuscript technically sound, and do the data support the conclusions?

Reviewer #1: Yes

Reviewer #4: Yes

3. Has the statistical analysis been performed appropriately and rigorously? 

Reviewer #1: Yes

Reviewer #4: Yes

4. Have the authors made all data underlying the findings in their manuscript fully available?

Reviewer #1: Yes

Reviewer #4: Yes

5. Is the manuscript presented in an intelligible fashion and written in standard English?

Reviewer #1: Yes

Reviewer #4: Yes

6. Review Comments to the Author

Reviewer #1: (No Response)

Reviewer #4: The revised manuscript was good written, designed and discussed. All comments of reviewers were addressed in the manuscript.

7. PLOS authors have the option to publish the peer review history of their article (what does this mean?). If published, this will include your full peer review and any attached files.

Reviewer #1: **Yes: **Mehmet Selim Çömez

Reviewer #4: No

---

## [Editor Report · Acceptance letter]

26 Aug 2024

PONE-D-24-10769R1 

PLOS ONE

Dear Dr. Pryambodho, 

I'm pleased to inform you that your manuscript has been deemed suitable for publication in PLOS ONE. Congratulations! Your manuscript is now being handed over to our production team.

Kind regards, 

on behalf of

Dr. Ramada Rateb Khasawneh 

Academic Editor

PLOS ONE